# Phenotypic diversity of *Methylobacterium* associated with rice landraces in North-East India

**Pratibha Sanjenbam, Radhika Buddidathi, Radhika Venkatesan, P. V. Shivaprasad, Deepa Agashe** *

National Centre for Biological Sciences, Tata Institute of Fundamental Research, Bangalore, India

* dagashe@ncbs.res.in

**Data Availability Statement:** Metadata for all samples is given in S1 Datasheet. Raw data for phenotypic assays and GCMS analysis of rice leaf surface are available on Figshare

## Abstract

The ecology and distribution of many bacteria is strongly associated with specific eukaryotic hosts. However, the impact of such host association on bacterial ecology and evolution is not well understood. Bacteria from the genus *Methylobacterium* consume plant-derived methanol, and are some of the most abundant and widespread plant-associated bacteria. In addition, many of these species impact plant fitness. To determine the ecology and distribution of *Methylobacterium* in nature, we sampled bacteria from 36 distinct rice landraces, traditionally grown in geographically isolated locations in North-East (NE) India. These landraces have been selected for diverse phenotypic traits by local communities, and we expected that the divergent selection on hosts may have also generated divergence in associated *Methylobacterium* strains. We determined the ability of 91 distinct rice-associated *Methylobacterium* isolates to use a panel of carbon sources, finding substantial variability in carbon use profiles. Consistent with our expectation, across spatial scales this phenotypic variation was largely explained by host landrace identity rather than geographical factors or bacterial taxonomy. However, variation in carbon utilisation was not correlated with sugar exudates on leaf surfaces, suggesting that bacterial carbon use profiles do not directly determine bacterial colonization across landraces. Finally, experiments showed that at least some rice landraces gain an early growth advantage from their specific phyllosphere-colonizing *Methylobacterium* strains. Together, our results suggest that landrace-specific host-microbial relationships may contribute to spatial structure in rice-associated *Methylobacterium* in a natural ecosystem. In turn, association with specific bacteria may provide new ways to preserve and understand diversity in one of the most important food crops of the world.

## Introduction

Microbial research has increasingly focused on the diversity and distribution of microbes in natural ecosystems. Multiple studies report substantial spatial structure across microbial populations and communities, separated by mere centimetres [1] to thousands of kilometres across continents [2]. Such spatial structure may reflect neutral or stochastic processes (including

(https://doi.org/10.6084/m9.figshare.11316269).
16S rRNA gene data is deposited in GenBank under
accession numbers: MN982765-MN982851 and
MN982724-MN982750.

**Funding:** This work was supported by: DA, Grant
DBT-358 NER/AGRI/24/2013 from the Department
of Biotechnology, India, and by the National Centre
for Biological Sciences. The funders had no role in
study design, data collection and analysis, decision
to publish, or preparation of the manuscript.

**Competing interests:** The authors have declared
that no competing interests exist.

dispersal limitation), and/or differential selection imposed by environmental factors [2], including association with eukaryotic hosts. For instance, local adaptation to specific host species or genotypes can drive substantial divergence across bacteria [3], perhaps best exemplified in plant pathogens [4]. Association with specific hosts can thus have large impacts on bacterial genomes [5], as well as functional traits such as those reflecting generalist vs. specialist life histories (e.g. reviewed in [3,4]). Host association is thus an important context to understand the diversity and distribution of bacteria, because it generates unique selection pressures compared to abiotic environmental factors. Additionally, strong structure in host populations themselves (e.g. due to limited dispersal or local adaptation to environmental conditions) may further enhance population structure in their bacterial associates.

Plant-associated bacteria–especially those occupying aerial niches (the "phyllosphere")–thus represent attractive systems to understand the processes and selection pressures that shape natural microbial populations. The phyllosphere is estimated to harbour nearly $10^7$ bacterial cells/cm$^2$ of leaf surface, with both biotic and abiotic factors influencing bacterial communities (reviewed in [6,7]). For instance, leaf exudates may include antimicrobials as well as nutrients; and colonization of leaf surfaces exposes bacteria to UV radiation and desiccation. These factors may impose selection that drives the evolution of bacterial traits such as aggregation, deployment of efflux pumps, and the production of protective pigments and biosurfactants [7]. Indeed, the composition of phyllosphere bacterial communities–typically dominated by Alphaproteobacteria and Gammaproteobacteria [8]–suggests a role for selection, rather than stochastic community assembly. However, factors shaping the populations of these abundant, typically non-pathogenic phyllosphere colonizers are not well understood. In particular, phenotypic diversity in natural microbial populations remains poorly explored. This is partly because it is difficult to quantify microbial phenotypic variation in nature, and to test for causal factors that shape this diversity [9]. Hence, the relative impact of abiotic vs. biotic factors in driving the diversity and distribution of plant-associated bacteria remains unclear.

Here, we addressed this problem by testing the impact of various geographical and host-related factors in shaping phenotypic variation in a genus of plant-associated bacteria. We analysed the distribution and functional divergence of rice-associated methylotrophic Alphaproteobacteria from the genus *Methylobacterium*. Bacteria from this genus are distributed globally and are abundant in the phyllosphere of many plant species, including rice [10]. This dominance of the phyllosphere is attributed to their unique ability to use methanol–released through plant stomata–as a sole carbon source [11]. Thus, the association with plants is thought to impose strong selection for methanol use in this genus. Indeed, proteins relevant for methanol use are very abundant during phyllosphere colonization [10,12,13]. In turn, colonizing *Methylobacterium* strains can enhance plant fitness. For instance, *Methylobacterium* colonizing the leaves of wheat plants increase seed germination rates by producing cytokinins [14]. Some *Methylobacterium* strains also produce auxin [15] and vitamin B12 [16], which may enhance plant growth. Additionally, *Methylobacterium oryzae* isolated from the phyllosphere of a rice variety in Korea is proposed to stimulate plant growth by using 1-aminocyclopropane 1-carboxylate (ACC)–a precursor of the plant stress hormone ethylene, which inhibits plant growth–as a nitrogen source [17]. Such mutualisms between rice plants and their associated *Methylobacterium* strains could potentially shape bacterial diversity and distribution in nature. Indeed, a recent study of ~3000 rice accessions from China and the Philippines found largely conserved leaf microbial communities and bacterial metabolism [18], suggesting that rice hosts may impose strong selection on bacterial communities. Thus, rice-associated *Methylobacterium* offer an exciting model to understand bacterial ecology and phenotypic diversity in nature, particularly in the context of host association.

We sampled 91 distinct epiphytic *Methylobacterium* isolates from the phyllospheres of 36 traditionally-grown rice landraces from geographically isolated locations in North-East (NE) India. These landraces have been cultivated by local communities for generations; typically, a given landrace is cultivated only in a restricted region and may be locally adapted (hence, we describe these as 'landraces' rather than 'cultivars', which are selected for a specific trait by breeders). As a result, there is substantial genetic diversity across rice landraces [19], potentially imposing divergent selection on their bacterial associates. In addition, the relatively high level of isolation between agricultural communities could also promote bacterial divergence via genetic drift and/or dispersal limitation. To determine whether association with distinct landraces could shape bacterial phenotype, we quantified the ability of our *Methylobacterium* isolates to use a panel of carbon sources. We found substantial metabolic variability across isolates, which was largely determined by the host rice landrace identity and not by sampling location or bacterial taxonomic classification. Importantly, rice-associated *Methylobacterium* isolates had distinct carbon use profiles compared to strains isolated from grasses and commercial rice varieties sampled from the same fields. Laboratory experiments showed that some rice landraces gain an early growth advantage from their specific *Methylobacterium* colonizers, although sugar exudates on rice leaves do not seem to determine bacterial colonization. Our work thus constitutes an important step towards understanding the role of host plant association in driving spatial structure and metabolic divergence in an important and dominant genus of phyllosphere bacteria.

## Materials and methods

### Media and growth conditions for *Methylobacterium* spp.

We used Hypho minimal medium [20] with the following composition. P solution: dipotassium phosphate ($K_2HPO_4.7H_2O$) 33.1g/L and sodium dihydrogen phosphate ($NaH_2PO_4.H_2O$) 25.9g/L; S solution: ammonium sulphate ($(NH4)_2.SO_4$) 5g/L and magnesium sulphate ($MgSO_4.7H_2O$) 2 g/L (Fisher Scientific); metal mix 1mL/L, pH 7. As appropriate, we supplemented it with 120mM filter-sterilized methanol (Fisher Scientific) as the sole carbon source, or other carbon sources (described below). We incubated cultures at 30˚C for 48h (liquid; with shaking) or 5 days (agar plates), unless mentioned otherwise. For long-term storage, we made glycerol stocks by mixing 500μL of liquid culture with 500μL of 50% glycerol (Merck) (in sterile water), and stored at -80˚C.

### Isolating *Methylobacterium spp*. from the phyllosphere of rice landraces and other sources

We first tested various parts of rice plants from fields near Bangalore, and determined that the flag leaf and stem have the highest density of *Methylobacterium*. We therefore sampled flag leaf and stem tissues of 36 distinct traditional rice landraces from geographically isolated locations in North-East India, focusing on the states of Arunachal Pradesh (Ziro district, East Siang district and West Siang district) and Manipur (Imphal East district and Thoubal district). We isolated *Methylobacterium* between September–October 2016 (Supplementary S1 Datasheet/S1 Table), when rice plants were at the heading (reproductive) stage (Fig 1). To test for temporal variation in *Methylobacterium* colonization, we again sampled seven landraces from Manipur in September 2017 (S2 Table). We sampled three randomly chosen plants per landrace, cutting out a 3–4 cm piece of the stem and the flag leaf from each plant and storing it temporarily in a clean plastic bag. Within a few hours, we imprinted (pressed) the stem and leaf samples onto agar plates containing Hypho minimal medium and 50mg/mL cycloheximide (Sigma-Aldrich)

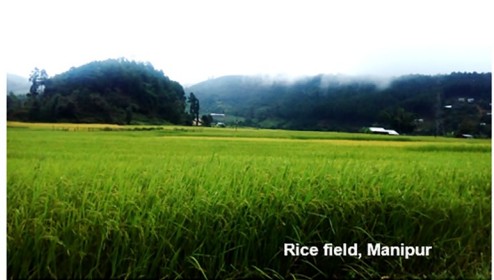
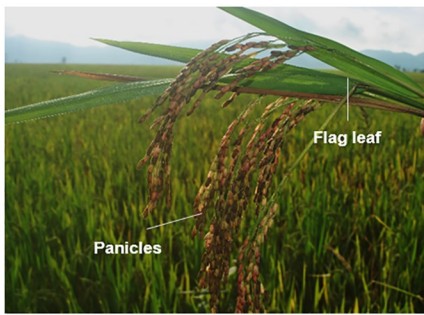

1. Collect flag leaf & stem samples

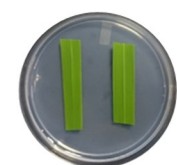
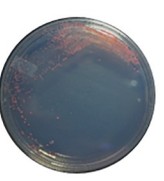
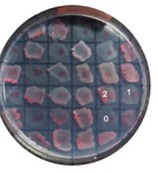

0 = No growth
1 = Poor growth
2 = Good growth

32 isolates/plate

2. Imprint on Hypho agar plate with 0.5% methanol

3. Pick pink colonies with distinct morphology

4. Obtain pure culture & sequence16S rRNA

5. Test for growth on 11 carbon sources

**Fig 1. Overview of sampling and experimental design.** We sampled phyllosphere *Methylobacterium* from stems and flag leaves of heading stage plants of 36 traditionally cultivated rice landraces (20 from Arunachal Pradesh and 16 from Manipur).

to prevent fungal growth. Since the field sites were remote and inaccessible, we imprinted plates on a clean work surface, inside a customized clear portable enclosure sterilized with UV radiation (S1 Fig). We incubated imprinted agar plates at room temperature for 5–6 days until we observed characteristic, pink *Methylobacterium* colonies. We isolated and sub-cultured morphologically distinct *Methylobacterium* colonies per landrace, to obtain pure cultures for a total of 250 isolates (see Supplementary S1 Datasheet for details). When plates had low colony density, we picked 4 colonies; when the density was high, we picked a maximum of 8 colonies.

To determine whether *Methylobacterium* strains associated differentially with the rice phyllosphere vs. other potential substrates, we isolated *Methylobacterium* from rice seed surfaces (4 landraces, S3 Table); the phyllosphere of grasses growing in rice fields (Chingarel site, Manipur, 2017); a recently developed green revolution line of rice (HYV701, Chingarel, Manipur, 2018) referred to as "commercial rice" hereafter; and from paddy field soil from fallow fields (in April 2016) and during rice cultivation (October 2016) (S3 Table). We sampled the phyllosphere of grasses and commercial rice as described above for landraces. For more extensive comparison of seed vs. phyllosphere colonizers, we also sampled the seed surface of 5 rice landraces from Arunachal Pradesh (S4 Table). To sample seed epiphytes, we placed 15 seeds of each landrace in 5mL Hypho broth with methanol, and incubated overnight at 30˚C. The next day, we plated serial dilutions of the broth on Hypho agar plates with methanol. For each landrace, we picked 4 colonies that had similar morphology. To sample soil *Methylobacterium*, we suspended 1g soil in 10mL PBS (Sodium chloride (NaCl) 8g/L, Disodium hydrogen phosphate ($Na_2HPO_4$) 1.44g/L, Potassium chloride (KCl) 200mg/L, Potassium dihydrogen phosphate ($KH_2PO_4$) 240mg/L (Fisher Scientific); pH 7.4) and plated serial dilutions of the supernatant on Hypho agar plates with methanol. We selected a total of 8 colonies from soil samples.

## Assigning *Methylobacterium* isolates to major taxonomic groups

For each isolate, we extracted genomic DNA (Wizard^R Genomic DNA Purification kit, USA) using 48h grown cultures inoculated from glycerol stocks. Using the primers 27F (5′ AGAGTTTGATCCTGGCTCAG 3′) and 1492R (5′ ACGGCTACCTTGTTACGACT 3′) [21], we amplified ~1400 bp of the 16S rRNA gene, and sequenced the PCR-purified product by Sanger sequencing. This region includes the V1-V6 hypervariable regions of the 16S rRNA gene. We used 16S rRNA sequences to create a multiple alignment in MEGA 7 [22], and constructed a neighbour-joining tree with 1000 bootstraps, including closely related reference strains identified using NCBI BLAST (see supplementary S1 Datasheet). We visualized the tree using the Interactive Tree of Life [23].

## Testing carbon utilization ability of *Methylobacterium* isolates

We tested the ability of each *Methylobacterium* isolate to use eleven different carbon sources (5mM of glucose, fucose, sucrose, xylose, arabinose, sodium tartarate, betaine, sorbitol, succinate, methylamine, or methanol; all from Sigma-Aldrich). We allowed each isolate to grow for 48h on Hypho plates with methanol, and patched a colony [24] on Hypho agar plates supplemented with one carbon source (filtered sterilized), with three biological replicates per isolate (i.e. three different colonies on three different plates). After 6 days of incubation, we scored the growth on each plate as follows: 0 = no growth, 1 = partial growth, and 2 = good growth (Fig 1). Separately, we tested the reliability of the growth assay by repeating it for a randomly chosen subset of 30 isolates on four carbon sources. For glucose, xylose and sucrose, results of the two assays were very consistent (0–6% mismatches), but growth on fructose was more variable (~16% mismatches) (S5 Table).

## Identifying potential cases of repeated sampling of isolates

Because the 16S rRNA gene provides limited resolution, we could have repeatedly sampled the same strain from a given landrace. To identify such cases, we used carbon source growth profiles (described in the next section; see supplementary information for details). For each landrace, we identified isolates with ≥ 96% 16S rRNA sequence identity and identical growth profiles (across 11 carbon sources) as potential cases of repeated sampling of the same strain. For subsequent analyses, we removed these potential duplicates, resulting in a dataset of 91 phyllosphere isolates from samples collected in 2016. Similarly, we obtained 24 distinct isolates in 2017 from the focal landraces; 3 distinct isolates from paddy field soil; and 5 distinct isolates from seeds of focal landraces from Chingarel. Note that we only removed potential duplicates within each landrace (or source), leaving a set of "unique" strains from each landrace. We did not remove potential clones shared across landraces, since these could be biologically meaningful cases of shared colonizers.

## Measuring concentration-dependent growth on selected carbon sources

For a randomly chosen subset of 58 *Methylobacterium* isolates, we tested the ability to use three different concentrations (5mM, 15mM, and 25mM) of 5 carbon sources that we detected on the rice phyllosphere (fructose, glucose, sucrose, xylose, and methanol; see next section). As described above, we revived cultures from glycerol stocks and patched them onto Hypho agar plates supplemented with each carbon source, with three biological replicates per isolate. We scored growth as described above. From these results, we classified isolates as showing one of four possible growth patterns for each carbon source: no growth at any concentration;

concentration-independent growth (i.e. growth at all tested concentrations); growth only at higher concentrations; and growth only at lower concentrations.

## Quantifying free sugars exuded on rice leaf surfaces

To test whether the phenotypic variation observed in the growth of *Methylobacterium* on different carbon sources is correlated with the concentration of sugars secreted by different rice varieties, we quantified the sugars present on the rice leaf surface using GCMS analysis. We collected flag leaf samples for a subset of landraces (4 rice landraces from East Siang district in Arunachal Pradesh and 4 rice landraces from Imphal East district in Manipur) between September–October 2018. We also collected the leaf blade next to the flag leaf (hereafter "regular leaf") to estimate leaf-specific variation in the sugar exuded by each landrace. We immediately placed leaves in a test tube containing MS grade water. Each biological replicate consisted of 5 leaves (each from a different plant) placed in a single tube containing 20mL water. In this manner, we collected 3 biological replicates for each rice landrace. Our main aim was to collect the surface sugars on the leaves, which should be easily accessible for epiphytic *Methylobacterium*. Hence, we made sure that the part of the leaf where we had cut it away from the plant did not touch the water. We removed all leaves from the tube after 1.5h, during which surface sugars could dissolve in the water. We filter-sterilized the water to prevent microbial growth. After removing leaves from the tube, we scanned each leaf using a flatbed scanner and used Black-spot software [25] to calculate the rice leaf surface area that was dipped in water. Using this, we calculated the amount of sugar secreted per unit leaf area.

In the lab, we concentrated each sample using a rotary evaporator at 70˚C under vacuum. We dried aliquots of 100μL of each sample in a lyophilizer and derivatized by silylation. For this, we added 40uL of dry pyridine and 60μL of MSTFA (N-methyl-N-(trimethylsilyl)- tri-fluoroacetamide) to the lyophilized sample and heated the mixture at 60˚C for 1h. Within 24h, we analysed 1μL of the silylated sample using a 6890C gas chromatograph (Agilent) interfaced with a 7000C mass selective detector (GC–MS). We used a DB5-MS (Agilent) capillary column (30m × 0.25mm I.D. and film thickness 0.25μm) with He as the carrier gas at a constant flow rate of 3mL min$^{-1}$. We maintained the injector and MS source temperatures at 280˚C and 300˚C, respectively. The column temperature program started at 60˚C with a hold time of 1min, followed by an increase to 190˚C at 20˚C min$^{-1}$ and a hold for 1min, followed by a temperature increase of 5˚Cmin$^{-1}$ to 280˚C, held for 3mins. We analysed samples in a split ratio of 5:1. We processed the data with HP-Chemstation software. We identified the sugars by comparison with the chromatographic retention characteristics and mass spectra of standards, reported mass spectra and the mass spectral library (NIST/EPA/NIH Mass spectral Library, Ver. 2.2, 2014). For quantification, we used calibration curves of external standards (glucose and fructose for monosaccharides; sucrose for disaccharides). Finally, we used the leaf surface area for normalization. To avoid background interference, we ran blanks in sequence after each set of samples.

## Testing for the impact of *Methylobacterium* on host plant growth

To test whether *Methylobacterium* isolates affected host plant fitness, we conducted two experiments: (1) Inoculating three rice landraces (Phou-ngang, Chakhao, and Gegong) with their respective associated *Methylobacterium* isolates (2) Inoculating a randomly chosen rice landrace (Ammo) with four different *Methylobacterium* isolates from other rice landraces (for both experiments, n = 3 plants/treatment/landrace). We first surface-sterilized 15 rice seeds of each landrace by washing with 70% ethanol (5mins), sterile water (10mins), and 1% sodium hypochloride (2mins); followed by multiple washes with sterile water (until foam generation

stopped). We dried seeds completely on a sterile Whatman filter paper. A day earlier, we revived *Methylobacterium* isolates in 5mL Hypho broth with 120mM methanol for 24h at 30˚C in a shaking incubator. We added the surface-sterilized seeds to the bacterial culture tube and incubated overnight at 30˚C with shaking. As a control, we soaked surface-sterilized rice seeds in sterile hypho broth with 120mM methanol. We transferred seeds from each landrace on separate sterile petri dishes (100mm diameter) lined with sterile filter paper, and incubated in a dark room for 5 days, adding sterile water as required to keep the filter paper moist. We transplanted 9 germinated seedlings from each treatment into pots (3 seeds per pot) containing 250g of autoclaved soil. We placed the pots inside a large outdoor insect cage (40cm X 40cm X 60cm), and watered them on alternate days. On the 7th day, we sprayed each plant with 1mL of the relevant bacterial culture ($OD_{600nm}$ = 0.4, inoculated 48h earlier from glycerol stocks), or sterile broth as a control. We periodically measured the height of each plant (7, 14, 21, and 40 days after the first inoculation) to estimate the rate of plant growth (as a proxy for plant fitness).

## Data analysis and visualization

We analysed and visualized data using R [26]. To visualize carbon use profiles of *Methylobacterium* isolates, we generated heatmaps using the function 'heatmap.2' in the package 'gplots' [27]. We tested the impact of location, elevation and rice landrace identity on variation in carbon use profiles using PERMANOVA (Permutational analysis of variance), using the function 'Adonis' in the package 'Vegan' [28]. To visualize clustering of *Methylobacterium* isolates based on carbon use profiles, we carried out canonical variance analysis (CVA) using the package 'Vegan' [28], and generated plots using the packages 'Cluster' [28] and 'BiplotGUI' [29]. We tested whether the observed phenotypic distance between *Methylobacterium* isolates is correlated with geographic distance between sampling sites, using the function 'Mantel.test' in the package 'Ape' [28]. We used the package 'Geosphere' [30] for estimating geographic distance between sites, using their latitude and longitude. Finally, we used Pearson's Chi-squared test for comparing *Methylobacterium* community composition across states, and paired t-tests for testing the effect of *Methylobacterium* on host plant fitness.

## Results

We obtained a total of 250 *Methylobacterium* isolates from the phyllosphere of 36 traditionally grown rice landraces from two states in North east (NE) India, namely Arunachal Pradesh and Manipur (Fig 1). On average, we sampled 4 isolates per landrace (range: 4 to 8; supplementary S1 Datasheet). For each isolate, we amplified and sequenced the 16S rRNA gene to assign taxonomic identity, and tested the ability of the strain to grow on a panel of carbon sources (see Methods; the results of these analyses are discussed below). Although bacterial colonization of plants depends on multiple factors [7], carbon metabolism is a key factor limiting bacterial growth and is therefore used to characterize bacterial species and strains. Plant-associated bacteria also have significantly more genes involved in carbohydrate metabolism, compared to non-plant-associated species in the same habitat [5]. Thus, we chose to determine carbon use profiles since these phenotypes are most likely to reflect selection associated with plant colonization. Based on 16S rRNA sequence similarity and carbon use phenotype, we identified 159 putative clones (i.e. potential cases of a strain being sampled repeatedly from a given landrace), and removed these from our dataset (see Methods for details). We conducted all further analyses with the remaining 91 distinct *Methylobacterium* isolates (S1 Table).

Based on the 16S rRNA neighbour joining tree, we assigned each isolate to one of 5 major clades: *M. komagatae* clade, *M. radiotolerans* clade, *M. nodulans* clade, *M. extorquens* clade,

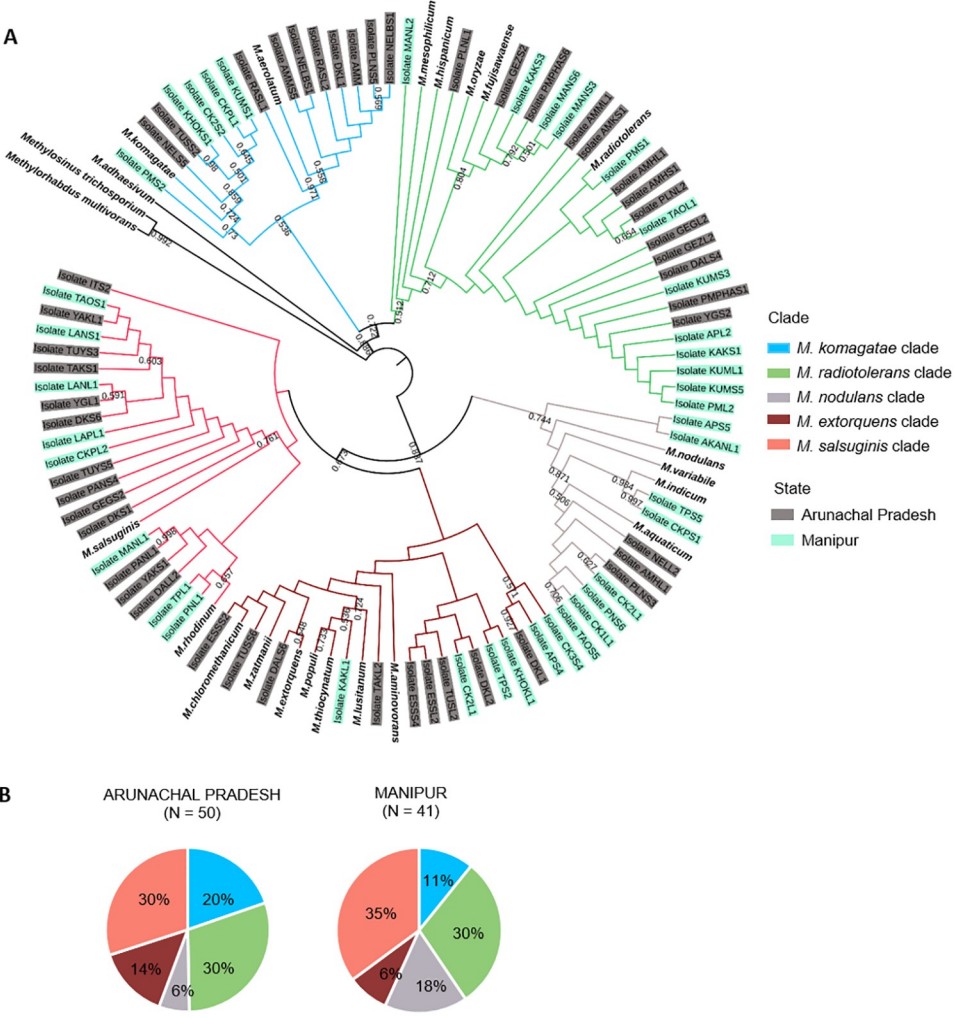

**Fig 2. Phylogenetic relationships between rice-associated *Methylobacterium* isolates.** (A) Neighbor-joining tree constructed from the 16S rRNA gene sequences of 91 distinct *Methylobacterium* isolates from rice landraces (colored by sampling state), and closely related reference strains (in bold). Bootstrap values ≥ 50% are indicated. We delineated five clades based on the tree (branches are colored as indicated). (B) Piecharts showing the clade composition of *Methylobacterium* isolates from each state.

and *M. salsuginis* clade (Fig 2A). Most isolates belonged to the *M. radiotolerans*, *M. nodulans* and *M. salsuginis* clades (Fig 2A; supplementary S1 Datasheet). Although representatives from all five clades were present in samples from both states (Fig 2A), the overall *Methylobacterium* community composition differed significantly across states, with relatively more isolates from the *M. nodulans* clade in rice landraces from Manipur (Pearson's Chi squared test, P = 0.01; Fig 2B). We note two caveats here: (1) The 16S rRNA sequence does not provide sufficient resolution to identify species definitively; and (2) Since our analysis of bacterial communities was not exhaustive (see Methods), this is not a comprehensive or quantitative dataset of the relative abundance of rice-associated *Methylobacterium* species. Nonetheless, our data provide an estimate of dominant *Methylobacterium* clades associated with different traditional rice varieties of NE India (Fig 2).

## Across spatial scales, variation in *Methylobacterium* carbon use is structured by host rice landrace

Across our set of 91 *Methylobacterium* isolates, we found substantial phenotypic variation in the use of 11 distinct carbon sources (Fig 3A), representing sugars commonly observed in leaf exudates of various plants [31–34]. We found that most isolates could use methanol, succinate, betaine, glucose, xylose, and arabinose; whereas very few could grow on sucrose, fucose, sorbitol, sodium tartrate, and methylamine. In general, the carbon use phenotype did not appear to correlate with the sampling state, the altitude of the sampling location, or the *Methylobacterium* clade to which the isolate belonged (Fig 3A). However, we explicitly tested the impact of a number of factors that might influence the observed phenotypic variability across isolates.

First, we tested the effect of geographical and environmental factors such as sampling state and elevation (altitude). Canonical variance analysis of the dataset showed that only ~4% of the phenotypic variation was explained by state (Fig 3B) and 7% by elevation (Fig 3C). Geographic distance between sampling sites was also uncorrelated with the phenotypic distance between *Methylobacterium* isolates (S2 Fig). In addition, we tested whether phenotypic variability was explained by taxonomic identity of isolates. However, clade identity explained only 5% of phenotypic variation across isolates (Fig 3D). Finally, we tested the impact of landrace, finding that 74% of the variation in carbon use was explained by host rice landrace (Fig 3E).

Prior work shows that sugar concentrations in leaf exudates can vary both within and across plants, under different environmental conditions [35]. Hence, the ability to show robust growth across different concentrations of a sugar may suggest strong selection to use that specific carbon source. Therefore, we next tested for variation in concentration-dependent growth, for a subset of 58 *Methylobacterium* isolates on 5 carbon sources. On methanol (thought to be a major source of selection on phyllosphere-associated *Methylobacterium* [36], 80% of the isolates showed consistently good and concentration-independent growth (S3A and S3B Fig), including at high concentrations of methanol that are toxic to most bacteria [37]. Although only about half the isolates could use fructose and glucose, growth on these substrates was also largely concentration-independent. In contrast, most isolates showed concentration-dependent growth on sucrose and xylose, and were unable to grow at high concentrations of sucrose or low concentrations of xylose. Next, we tested whether these concentration-dependent growth patterns could also be shaped by host landraces. Indeed, host landrace had the largest impact on the form of the concentration-dependent growth phenotype across the 5 carbon sources (explaining 67% of variation, Fig 3F), whereas sampling state, elevation and taxonomic identity each explained less than 28% of variation (S3C Fig). In other words, host landraces may influence not only the ability of *Methylobacterium* to use specific carbon sources, but also the form of the growth response.

Does host landrace identity also structure variation in bacterial carbon use profiles at smaller spatial scales, where we expect greater mixing of bacterial communities? In Arunachal Pradesh, landrace consistently explained >80% of variation in bacterial carbon use phenotype at different scales: within the state (Fig 4C), within a sampling region (Fig 4D) and within a single rice field where multiple landraces are cultivated (Fig 4E). Similarly, in Manipur, landrace identity explained 52–61% of bacterial carbon use across spatial scales (Fig 4F–4H). Even within the relatively small spatial scale of a single rice field, landrace identity explained a substantial fraction of bacterial phenotypic variation. Thus, our results suggest that colonization of the phyllosphere of rice landraces could play a major role in shaping bacterial metabolism.

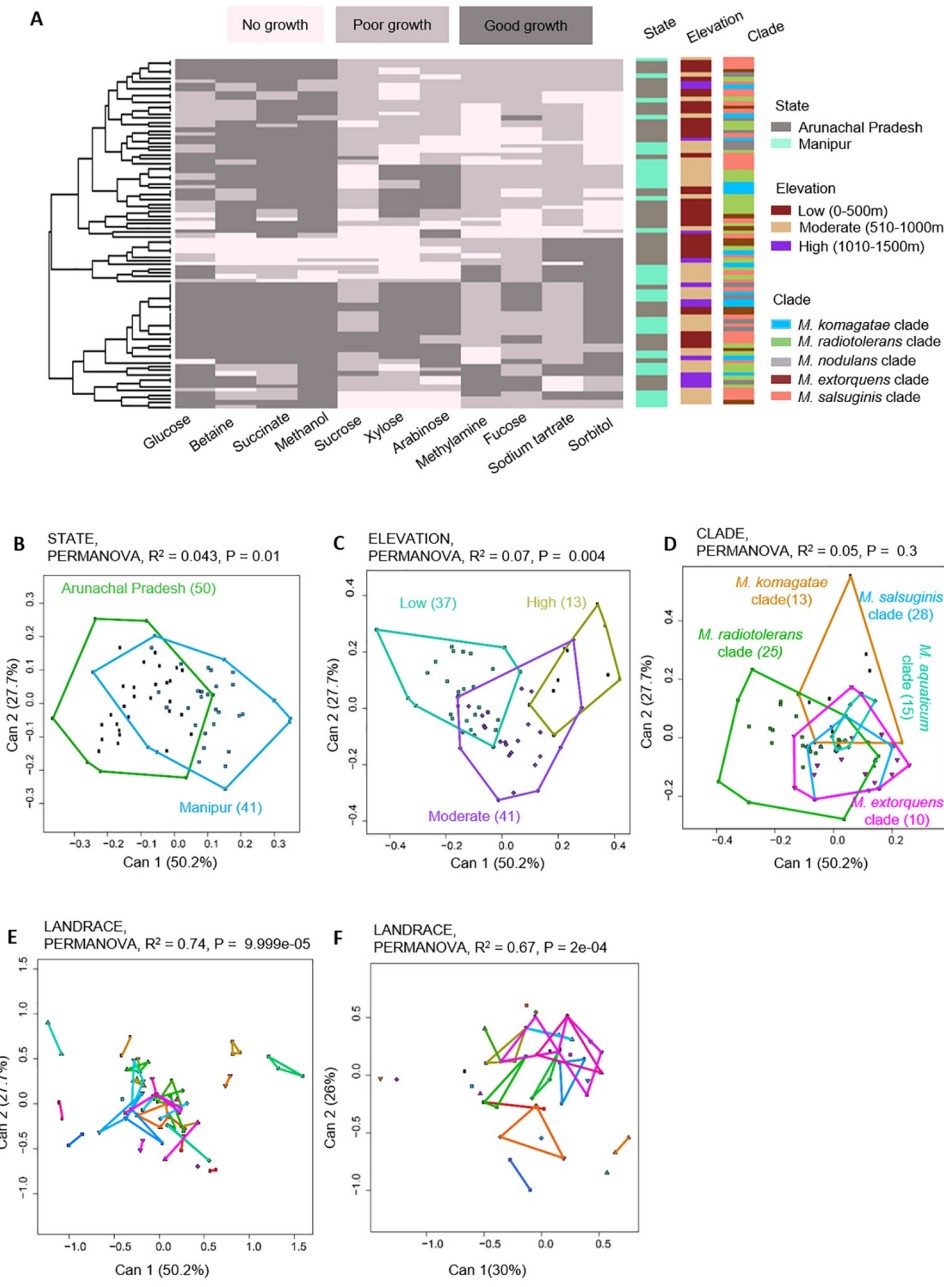

**Fig 3. Variability in carbon use profiles across rice-associated *Methylobacterium* isolates.** (A) A summary of the ability of 91 *Methylobacterium* isolates to utilize a panel of 11 carbon sources. Each row represents one isolate; each column represents a carbon source; and cells are colored based on qualitative growth of each isolate in each carbon source. Isolates are clustered (see dendrogram on the left) based on Euclidian distances and Ward clustering, given their carbon use phenotype. Columns to the right represent potential explanatory variables for the carbon use profile of each isolate, and are colored by sampling state, elevation, and *Methylobacterium* clade identity. (B–F) Canonical variance analysis biplots of carbon use profiles, showing the clustering of all 101 *Methylobacterium* isolates by sampling state (B), elevation (C), *Methylobacterium* clade (D), rice landrace (E-F). In panels B, C, E and F, each point represents an isolate, and numbers in parentheses indicate the number of isolates in each group. Axis labels indicate the proportion of variation explained. Convex hulls represent 95% confidence intervals.

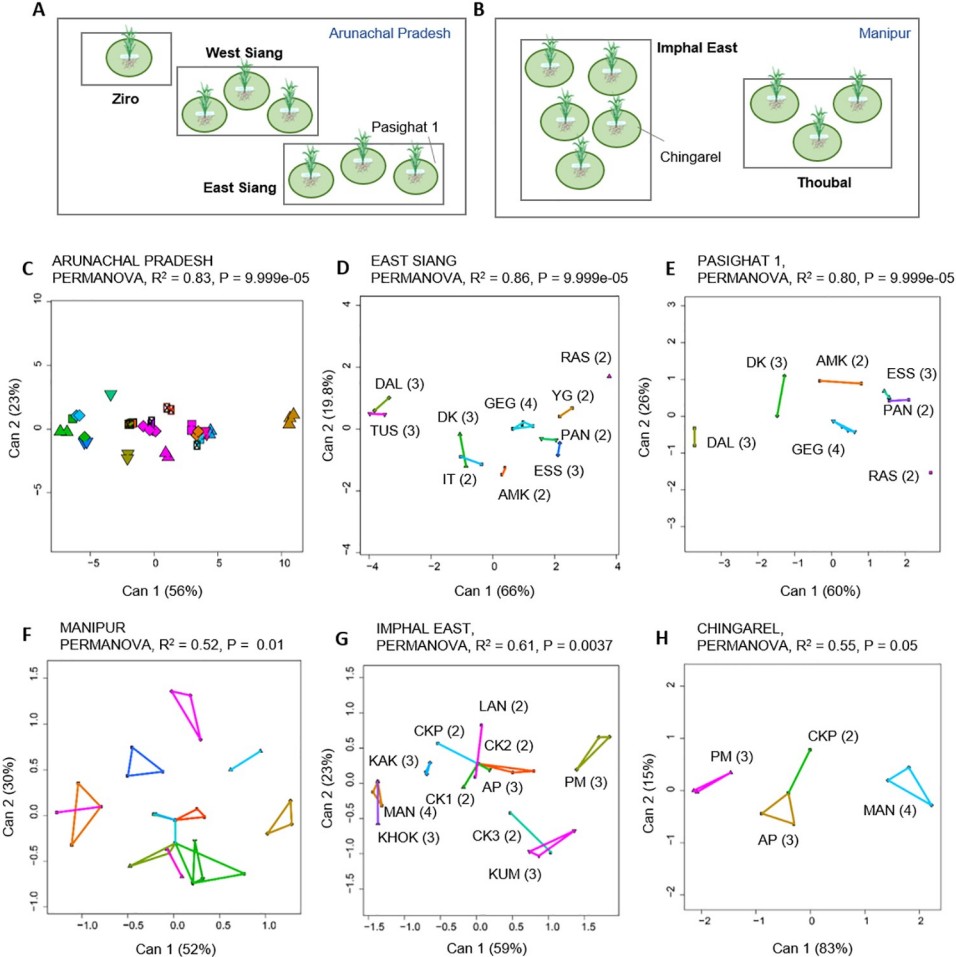

**Fig 4. Phenotypic variability of *Methylobacterium* across spatial scales.** (A–B) Schematic showing different regions (highlighted in bold) and field sites sampled in (A) Arunachal Pradesh, and (B) Manipur. Each circle represents a sampled rice field. The schematic is not to scale. (C–H) Canonical variance analysis biplots of carbon use profiles, showing the clustering of *Methylobacterium* isolates by landraces sampled (C) within Arunachal Pradesh, (D) within the East Siang region in Arunachal Pradesh, (E) within a single field (Pasighat 1) in Arunachal Pradesh; (F) within Manipur, (G) within the Imphal East region of Manipur, and (H) within a single field (Chingarel) in Manipur. Each point represents an isolate, and numbers in parentheses indicate the number of distinct isolates from each landrace (landrace names are abbreviated as indicated in S2 Table). In panels C and F, we omitted landrace names for clarity. In panels C–G, axis labels indicate the proportion of variation explained, and convex hulls represent 95% confidence intervals for each landrace (colors indicate different landraces).

## *Methylobacterium* strains specifically colonize the rice phyllosphere

The consistent signature of host identity on the spatial structure of bacterial populations suggested that *Methylobacterium* colonization may be strongly influenced by host-specific filtering of the local *Methylobacterium* community, reflecting host selection or competition between colonizing strains. Alternatively, colonization patterns may be driven largely by stochastic variability in bacterial dispersal at a finer spatial scale, with a minimal role for host identity. If so, we would expect that *Methylobacterium* from the rice phyllosphere would be similar to other isolates in the local environment. To explore this possibility, we sampled *Methylobacterium* from various other sources in the Chingarel field in Manipur: (1) the phyllosphere of grasses growing in the same field, (2) the surface of seeds from focal rice landraces,

(3) a commercial rice variety cultivated in the same field, and (4) field soil. Based on their taxonomic identity and carbon use profiles (as described above), we removed putative clones (see Methods) and obtained 4 distinct isolates from grasses; 5 isolates from seeds of four rice landraces; 4 isolates from the commercial rice variety; and 3 distinct isolates from field soil sampled between rice growing seasons (i.e. fallow field). Field soil collected during rice cultivation did not yield any *Methylobacterium* colonies; instead, we found bacteria from the genera *Rhodopseudomonas*, *Bradyrhizobium*, and *Mesorhizobium* (identified based on the 16S rRNA gene sequence).

Interestingly, we found that different *Methylobacterium* strains colonized different surfaces in the same field. While leaf imprints and seed surfaces from our focal landraces had abundant *Methylobacterium* colonies, grasses and the commercial rice variety produced very few colonies, indicating generally poor colonization (S4 Fig). Grass, commercial rice, landrace seed and soil samples also harboured less *Methylobacterium* richness (Fig 5A). For instance, we did not detect bacteria from the *M. nodulans* and *M. komagatae* clades in these samples; but both clades were well represented in rice landrace phyllospheres from the same field. *Methylobacterium* strains isolated from different sources also varied in their carbon use profiles (Fig 5B). For instance, all *Methylobacterium* isolates from grasses and commercial rice could grow on methylamine, whereas most isolates from landraces, seed surfaces, and soil could not use methylamine. Some of this phenotypic variability was explained by clade (14%; Fig 5C), but the largest fraction of variation was explained by the sampling source (37%; Fig 5D). Hence, we speculate that the *Methylobacterium* isolates that we obtained from traditional rice landraces are especially good colonizers of the phyllosphere of those rice varieties, and do not colonize grasses or commercial rice grown in the same field.

To explicitly test whether phyllosphere bacteria specifically colonize leaf and stem surfaces, we isolated *Methylobacterium* from seed surfaces of 10 of our focal landraces, and compared them with the respective *Methylobacterium* phyllosphere community. As observed at the Chingarel field, seed epiphytic communities appeared to be less rich than phyllosphere communities of *Methylobacterium*, although the difference was not significant (Pearson's Chi squared test, P = 0.28; Fig 6A; also see S5 Fig for results for each landrace). On the other hand, seed and phyllosphere epiphytes did not show major differences in their carbon use profiles (Fig 6B), although for many landraces, seed isolates tended to use fewer carbon sources than phyllosphere isolates (S5 Fig). Overall, only 7% of the variation was explained by sampling state; 10% by *Methylobacterium* clade; and 2% by the sampling source (phyllosphere vs. seed) (Fig 6C–6E). However, landrace identity explained 43% of variation in the carbon use phenotype (Fig 6F), suggesting that seed surface and phyllosphere communities were generally more similar within than across landraces.

Together, these results suggested that the phyllosphere of rice landraces is colonized by taxonomically and phenotypically different *Methylobacterium* communities compared to other available niches, including seed surfaces of rice landraces. These results indicate potential filtering during leaf colonization, and/or differential outcomes of strain competition on different niches available to the local bacterial community.

## *Methylobacterium* strains colonizing rice phyllosphere are temporally variable

If rice landraces select for particular *Methylobacterium* colonizers with specific metabolic capability, we would expect temporal consistency in the identity and/or carbon use profile of colonizing strains, depending on the target of host selection. To test this, we re-sampled seven landraces from Manipur in 2017, at the same stage of host plant growth. Isolates sampled in

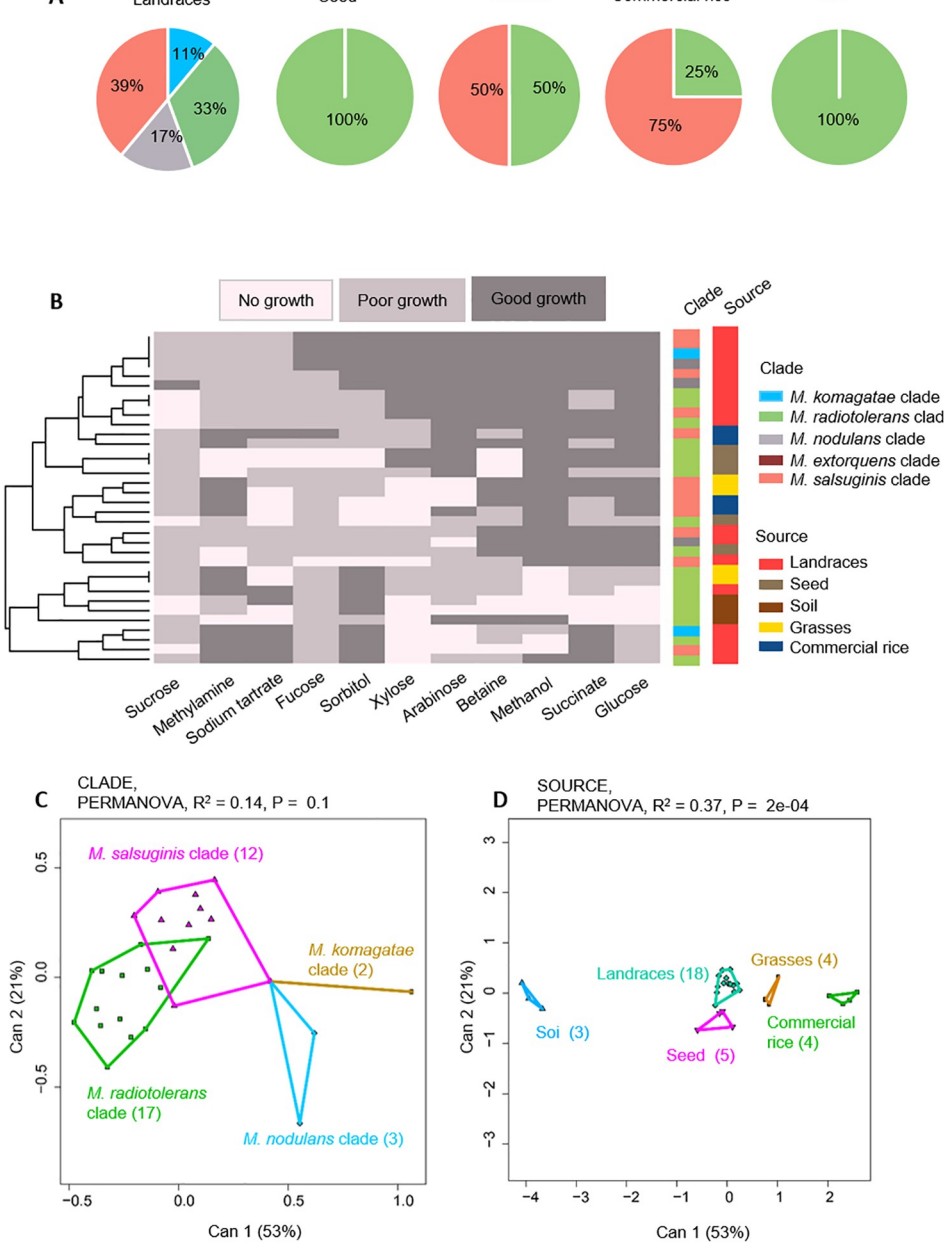

**Fig 5. Phenotypic variation of *Methylobacterium* from different sources.** (A) Piecharts showing *Methylobacterium* clade composition across rice landraces (sampled in 2016), seed surfaces of landraces, grasses, commercial rice, and soil collected from the same field. (B) A summary of the ability of 34 *Methylobacterium* isolates (from the sources described in panel A) to utilize a panel of 11 carbon sources. Each row represents one isolate; each column represents a carbon source; and cells are colored based on qualitative growth of each isolate in each carbon source. Isolates are clustered (see dendrogram on the left) based on Euclidian distances and Ward clustering, given their carbon use phenotype. Columns to the right represent potential explanatory variables for the carbon use profile of each isolate, and are colored by by source and *Methylobacterium* clade. (C-D) Canonical variance analysis biplots of carbon use profiles, showing the clustering of all 34 *Methylobacterium* isolates by (C) *Methylobacterium* clade, and (D) source. Each point represents an isolate, and numbers in parentheses indicate the number of isolates in each group. Convex hulls represent 95% confidence intervals.

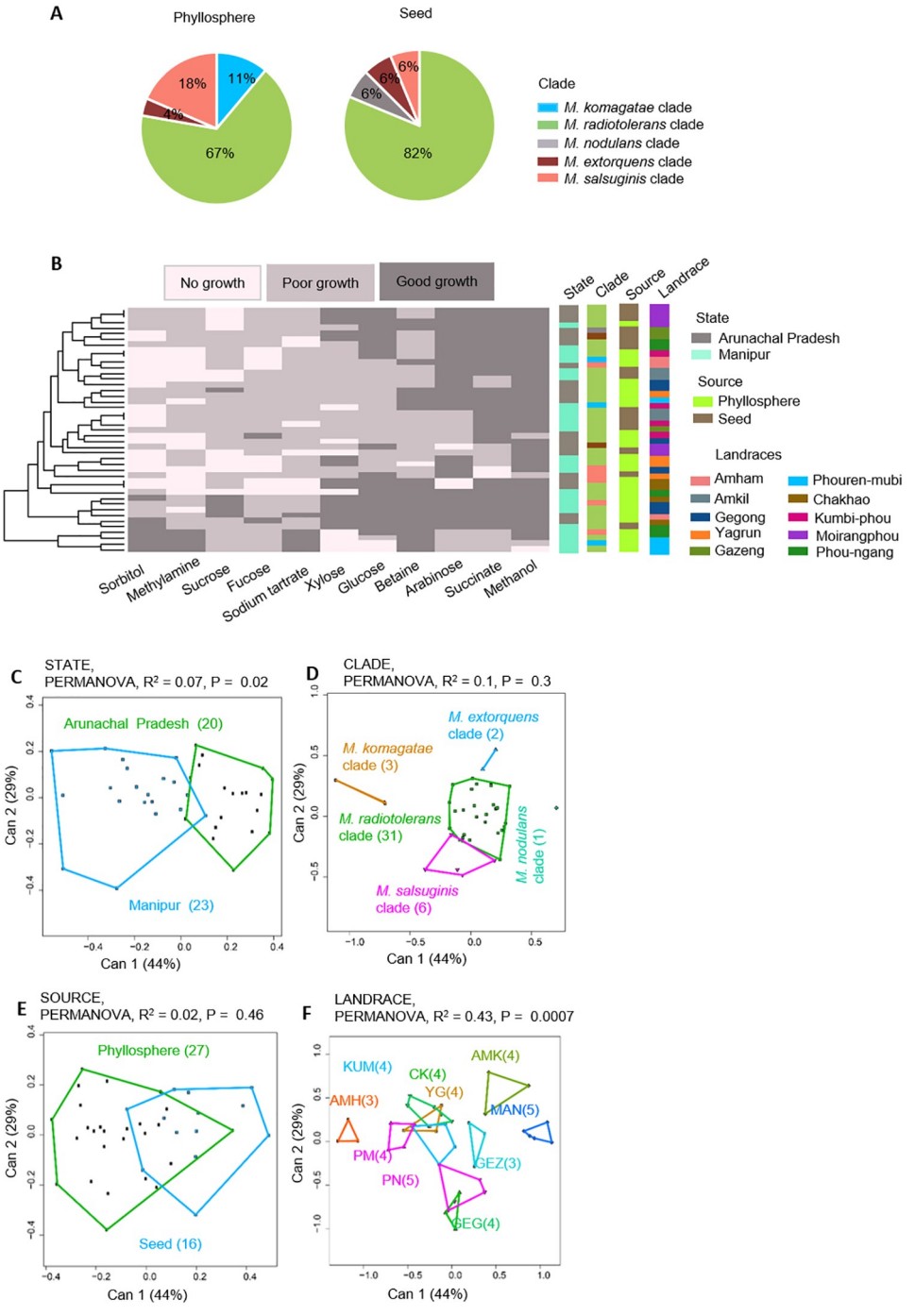

**Fig 6. Phenotypic variation in the *Methylobacterium* isolates from phyllosphere and seed.** (A) Piecharts showing *Methylobacterium* clade composition from phyllosphere vs. seed surface. (B) A summary of the ability of 43 *Methylobacterium* isolates (from sources described in panel A) to utilize a panel of 11 carbon sources. Each row represents one isolate; each column represents a carbon source; and cells are colored based on qualitative growth of each isolate in each carbon source. Isolates are clustered (see dendrogram on the left) based on Euclidian distances and Ward clustering, given their carbon use phenotype. Columns to the right represent potential explanatory variables for the carbon use profile of each isolate, and are colored by state, source, *Methylobacterium* clade, and landrace. (C-F) Canonical variance analysis biplots of carbon use profiles, showing the clustering of all 43 *Methylobacterium* isolates by (C) state, (D) *Methylobacterium* clade, (E) source, and (F) landrace. Each point represents an isolate, and numbers in parentheses indicate the number of isolates in each group. Convex hulls represent 95% confidence intervals.

2016 also belonged to the same five clades as the 2017 isolates (S6 Fig). However, the *Methylobacterium* community composition was significantly different across years (Pearson's Chi-squared test, P = 0.015, Fig 7A). In 2017, the same landraces harboured a larger fraction of isolates from the *M. komagatae* and *M. extorquens* clades, and fewer isolates from the *M. radiotolerans* clade. The degree of variation in strain identity also differed across landraces: some rice landraces such as Chakhao, Kumbi-phou and Moirang-phou hosted similar clades across years, while the other four varieties showed dramatic shifts in community composition (S7 Fig). These results do not support the hypothesis that rice landraces are consistently colonized by specific *Methylobacterium* strains.

The lack of support for selection of specific strains is perhaps not surprising in light of the lack of a strong taxonomic signature in *Methylobacterium* carbon use profiles (Fig 3E). Thus, it is plausible that from the pool of locally available *Methylobacterium*, any strain with the appropriate metabolic capacity could colonize a given landrace, with strain identity varying stochastically across years. If so, we would expect that despite temporal variation in the identity of colonizing bacteria, across years we should observe similar carbon use profiles for strains colonizing each landrace. However, the carbon use profiles of *Methylobacterium* isolates also differed across years (Fig 7B), with variable degrees of temporal consistency across landraces (S7 Fig). Overall, combining data from both years, rice landrace still explained the largest fraction of variation (30%, Fig 7C); but a substantial amount of variation (23%) was also explained by sampling year (Fig 7D).

Together, these results suggested that NE rice landraces are colonized by a set of related, yet metabolically distinct *Methylobacterium* isolates across years. Whereas some landraces may select for specific isolates or carbon use profiles, all landraces do not show evidence of such selection. Thus, the phenotypic variability of *Methylobacterium* isolates associated with NE rice landraces appears to be shaped by both landrace and the physical environment, although the precise causal environmental factors remain unknown.

## Sugar availability on rice leaves does not explain phenotypic variation across *Methylobacterium*

As discussed above, despite the strong impact of rice landrace identity, the temporal inconsistency in the carbon use of rice-associated *Methylobacterium* isolates was puzzling because it suggested a lack of direct selection for specific bacterial traits. However, this contradiction may be explained by the limitations of our phenotypic assay: host rice landraces may select for bacterial traits that are unrelated to the use of carbon sources that we assayed. To test this hypothesis, we used GC-MS analysis to quantify the sugars exuded on field-collected leaf surfaces of four landraces each from Manipur and Arunachal Pradesh. We sampled flag leaves as well as "regular" leaves (the leaf blade closest to the flag leaf). Flag leaves are directly relevant to our study because we sampled *Methylobacterium* from flag leaves; and measuring sugar production on other leaves allows a comparison across leaf types from the same plant. A positive correlation between available flag leaf sugars and bacterial sugar use capability for each landrace would indicate potential host-imposed selection on bacterial carbon use.

We found that different landraces produced different sugars in distinct concentrations on their leaf surfaces (Fig 8A). Overall, fucose, fructose, glucose, and sucrose were abundant and commonly found in leaf exudates; but we could not detect arabinose, galactose, mannitol, mannose, sorbitol, and xylose. In six landraces, glucose was the most abundant sugar; whereas in two other landraces, fucose was most abundant. In most cases, the flag leaf had lower sugar concentrations compared to the regular leaf, potentially reflecting the transport of sugars from the flag leaf to panicles for grain filling [38]. Thus, nutrient availability on the leaf surface

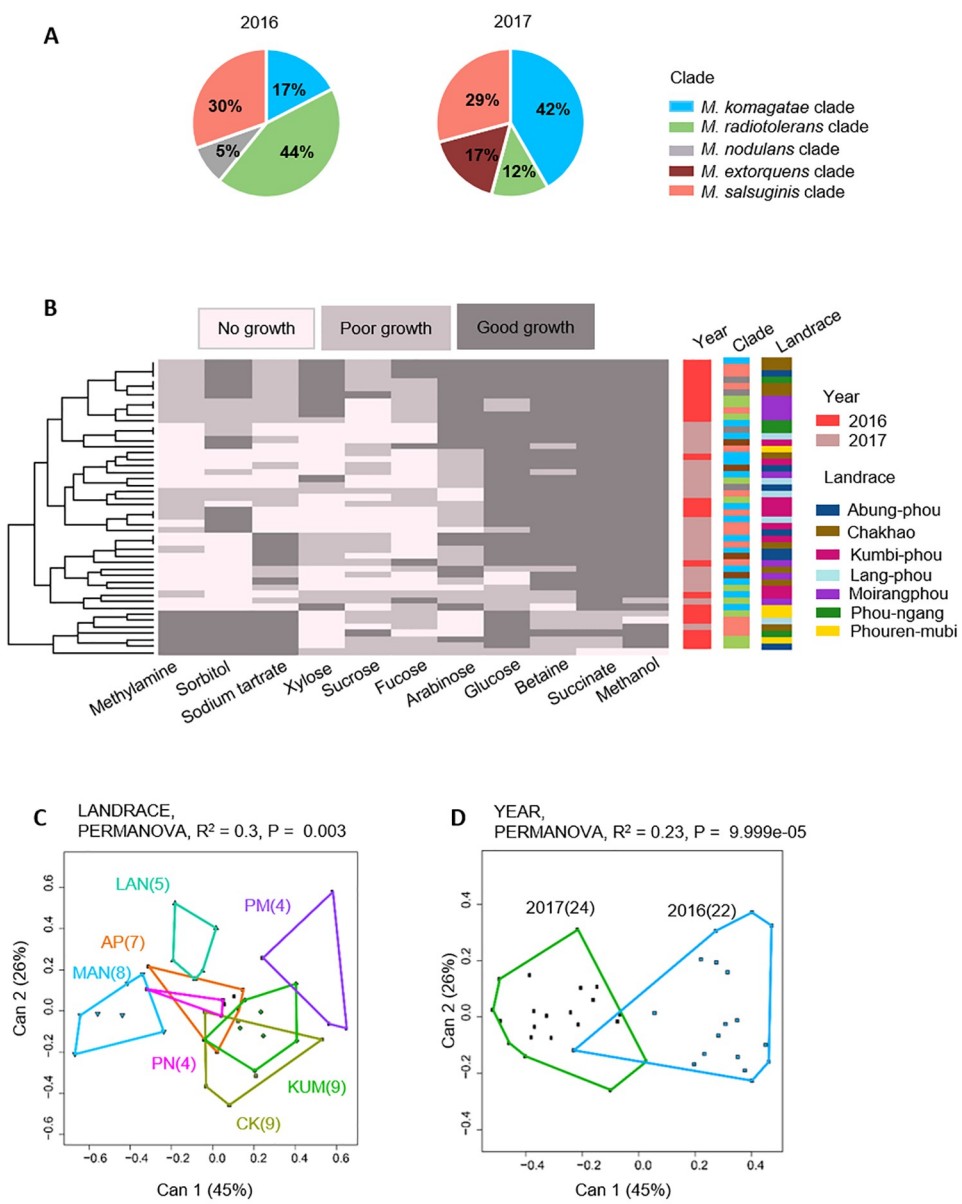

**Fig 7. Temporal variation in the composition and carbon use profiles of rice-associated *Methylobacterium* isolates.** Of the 36 rice landraces sampled in 2016, we re-sampled seven landraces from Manipur in 2017. (A) Piecharts summarizing the clade composition of *Methylobacterium* isolates from focal landraces across years. Separate data for each landrace are shown in S7 Fig. Isolates are clustered (see dendrogram on the left) based on Euclidian distances and Ward clustering, given their carbon use phenotype. Columns to the right represent potential explanatory variables for the carbon use profile of each isolate, and are colored by year, clade, and landrace. (B) A summary of the carbon use profiles of *Methylobacterium* isolates from focal landraces across years; plot details are as described in Fig 2. (C–D) Canonical variance analysis biplots of carbon use profiles, showing the clustering of *Methylobacterium* isolates by (C) landrace and (D) year. In panels C–D, each point represents an isolate, and numbers in parentheses indicate the number of isolates in each group. Convex hulls represent 95% confidence intervals.

varied both across landraces and across leaf types of each landrace. Next, we asked whether the variability in available sugars was correlated with the ability of colonizing *Methylobacterium* to use the respective sugars. Surprisingly, although all eight focal landraces exuded fucose, their phyllosphere *Methylobacterium* strains could not use fucose for growth (Fig 8B). Similarly,

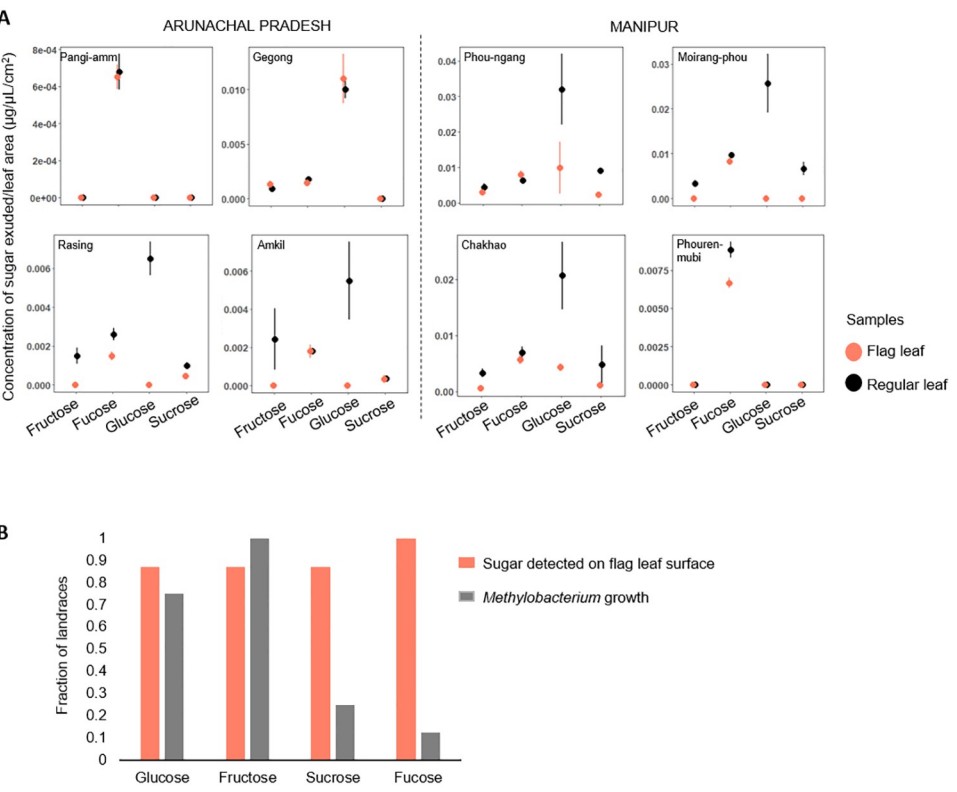

**Fig 8. Sugar availability on rice leaf surface does not explain carbon use profiles of *Methylobacterium* isolates.** (A) Plots show the mean concentration (±SE) of sugars detected on rice leaf surfaces of focal landraces from Arunachal Pradesh and Manipur (n = 3 samples / leaf type / landrace. (B) Bar plot shows the proportion of landraces that produce a specific sugar, and the ability of associated *Methylobacterium* strains to use each sugar.

although many landraces produced detectable levels of sucrose, colonizing *Methylobacterium* could not use sucrose as the sole carbon source. On the other hand, most bacteria were able to use both fructose and glucose; even when fructose was not detected on the host leaf surface. Finally, we note that sugars such as arabinose, xylose, and sorbitol–which could be used for growth by some isolates–were not detected in any of our leaf samples (Fig 8A; S8 Fig).

Overall, our results are generally inconsistent with the hypothesis that sugar availability in the phyllosphere drives variation in carbon use profiles of *Methylobacterium* isolates. Thus, *Methylobacterium* colonization success on the rice phyllosphere likely depends on traits that we did not measure, and that are uncorrelated with sugar use. Alternatively, the observed bacterial metabolic variation may be driven by stochastic processes (e.g. environmental fluctuations) or constraints imposed by past evolutionary history (e.g. tradeoffs, or the opportunity for horizontal gene transfer of genes relevant for carbon use).

### The impact of colonization by *Methylobacterium* varies across landraces

Our results suggested the possibility that rice-*Methylobacterium* mutualism may contribute to the spatial structure in bacterial populations in NE India. To experimentally test whether such a mutualism is possible, we conducted two short-term laboratory experiments. First, we inoculated rice seedlings of three different landraces with their respective colonizing *Methylobacterium* strains. As a control, we inoculated plants with bacterial growth medium. In the first experiment, seeds from the Gegong landrace did not germinate, so we obtained results for

only two landraces. We found that *Methylobacterium* inoculation increased the growth of one landrace (Phou-ngang), but not the other (Chakhao) (Fig 9A). At the end of the experiment, we sequenced the 16S rRNA gene of *Methylobacterium* isolates from Phou-ngang plants, confirming that the phyllosphere was indeed colonized by the inoculated *Methylobacterium*. Hence, Phou-ngang may obtain a growth benefit from its associated *Methylobacterium*, making a mutualism plausible. To test whether the benefit to rice plants is specific or general, in a second experiment we separately inoculated a randomly chosen rice landrace (Ammo) with one of four *Methylobacterium* strains isolated from other landraces. In all cases, plant growth was no different from control plants treated with bacterial growth medium (Fig 9B). Together, these experiments suggest that some rice landraces may have a mutualistic and specific relationship with their phyllosphere-colonizing *Methylobacterium* strains.

## Discussion

Rice is one of the world's most important food crops, with an annual production of ~14 million tonnes that feeds millions of people [39]. In recent years, we have increasingly realized the value of traditional rice varieties (rice "landraces"), which encompass distinct genetic diversity and nutritional content [40], and can be highly robust to environmental fluctuations [41]. The north-eastern region of India–one of the world's biodiversity hotspots–is also home to hundreds of rice landraces cultivated in relatively isolated and remote areas of this region [42]. These landraces were selected for diverse phenotypic traits (e.g. aroma, grain colour and texture) by local communities, and the combination of divergent selection and limited gene flow presumably led to substantial genetic diversity across landraces [40,43,44]. We predicted that this diversity across rice varieties might also lead to significant spatial structure in rice-associated *Methylobacterium*, either by directly imposing differential selection on the bacterial populations or facilitating their divergence by restricting gene flow. Overall, our results are consistent with this broad hypothesis. First, across 91 distinct *Methylobacterium* strains isolated from the phyllosphere of 36 landraces, we found that landrace identity consistently explained a large fraction of variation (~70%) in bacterial ability to use a panel of 11 carbon sources. In contrast, other variables such as bacterial taxonomy, distance between sampling sites or site altitude were poor predictors of bacterial carbon use. Second, grasses and commercial rice varieties grown in the same area harboured very small *Methylobacterium* communities with distinct composition and carbon use profiles, suggesting landrace-specific bacterial colonizers. Third, although landraces harboured different strains across two years of sampling, landrace identity still explained the largest fraction of variation in the dataset (~30%). Fourth, different landraces grown in the same site (in the same or adjacent fields) also harboured *Methylobacterium* with distinct carbon use profiles, indicating a stronger impact of host identity rather than divergent environmental variables or restricted gene flow on bacterial metabolism. Finally, our laboratory experiment showed that at least one landrace gains a growth advantage specifically from *Methylobacterium* that colonizes its phyllosphere; setting up the possibility of a host-bacterial mutualism and host-imposed selection on bacterial partners. Of course, much more work is necessary to determine the extent, nature, and mechanisms of potential divergent host-imposed selection in driving the observed phenotypic variation in rice-associated *Methylobacterium* populations.

Remarkably, host landrace identity consistently explained a large fraction of variation in *Methylobacterium* carbon use profiles. Prior work showed similar spatial structuring of *Methylobacterium* communities of *Medicago* and *Arabidopsis*, as a function of site and plant species [45]. For instance, *Methylobacterium* strains colonizing soybean can use methylamine for growth, whereas rice-associated isolates do not do so [46]; this study); this difference is

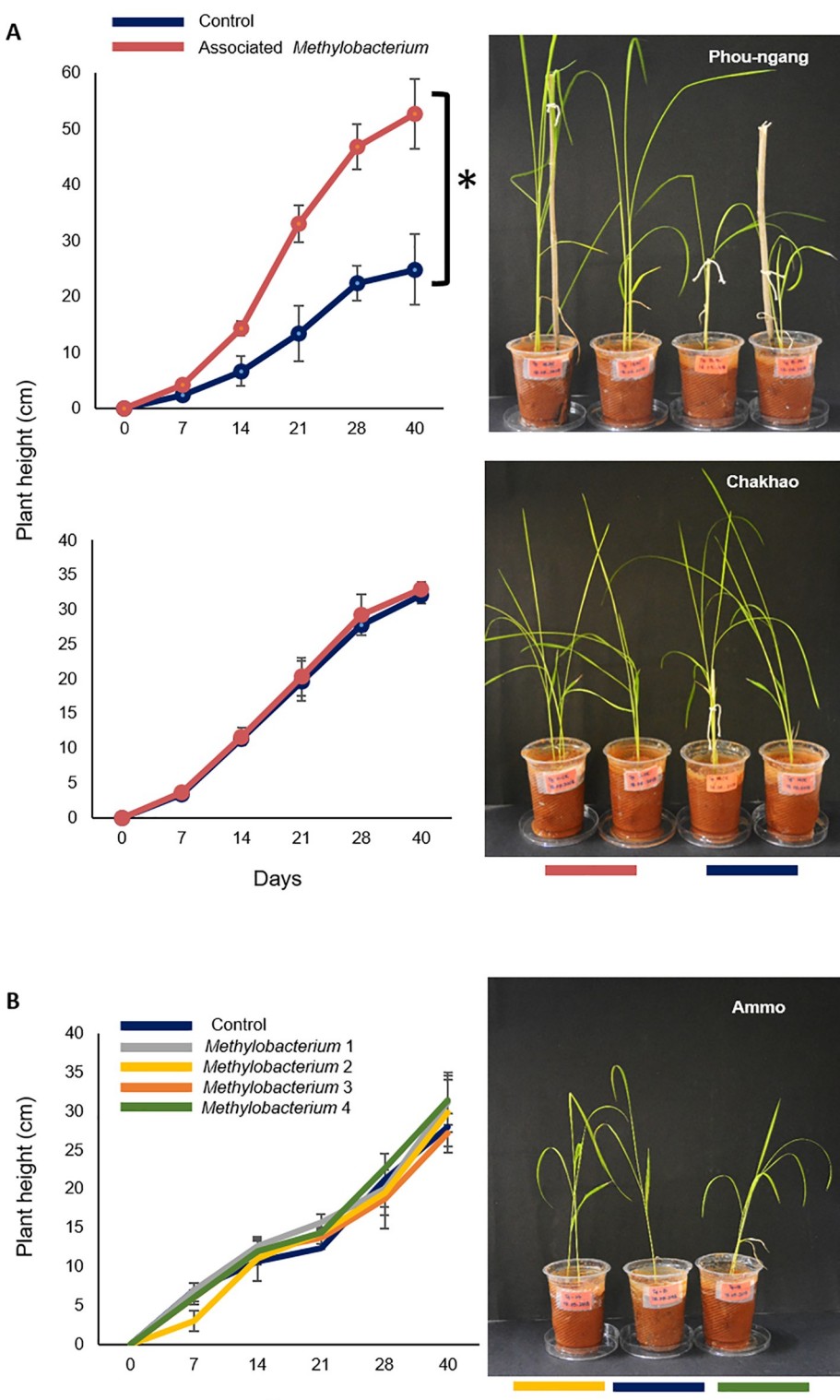

**Fig 9. The effect of *Methylobacterium* on early growth of host rice landraces.** Change in plant height (mean±SD) over time, measured for (A) two different landraces inoculated either with their associated *Methylobacterium* strain or with bacterial growth medium (control) (B) plants from the landrace Ammo inoculated either with *Methylobacterium* isolated from various (other) landraces, or with bacterial growth medium (control). In both experiments, n = 3 plants / treatment. Arrows indicate the day on which plants were inoculated with the respective *Methylobacterium* strain (or growth media, for controls). Pictures show plant height on the last day of the experiment (day 40).

proposed to reflect the plants' ability to produce methylamine. However, in our study, we did not find evidence for direct host-imposed selection on bacterial carbon use as a factor that could shape bacterial communities. This is surprising because prior work shows strong impacts of local environment (including host traits) on microbial communities. For instance, the carbon use profiles of ~500 natural yeast isolates were clustered by the substrate from which they were isolated [47]. More specifically, leaf chemistry is strongly correlated with phyllospheric microbial communities of tropical trees [48], and sugar concentration in host plant exudates affects colonization by epiphytic bacteria [49]. However, these effects may differ across bacteria. Although the net sugar secreted on *Arabidopsis* leaf surfaces was significantly reduced when inoculated with bacterial genera such as *Sphingomonas* and *Pseudomonas*, there was no reduction when inoculated with *Methylobacterium* [34]. Thus, *Methylobacterium* colonization may not depend on sugars. In our case, a primary filter on phyllosphere colonization by *Methylobacterium* could be placed by the ability to tolerate or use methanol and other single-carbon sources. After this filter, bacterial colonization might be determined not by sugar availability, but by the outcome of bacterial competition for other metabolites or nutrients, tolerance to UV exposure, or by host features such as leaf architecture [50], which we did not measure. For instance, a recent study showed that the auxotrophic *Methylobacterium* sp. OR01 requires plant-derived vitamin B5 (among other nutrients) to colonize the *Arabidopsis* leaf surface [51]. Thus, to better understand the ecology of *Methylobacterium* in the phyllosphere, we need to determine the role of host-specific factors that are critical for its colonization and growth.

A puzzling contradiction in our results is the observation that landraces harbour genetically and phenotypically distinct *Methylobacterium* isolates across years. Such temporal variation is common in natural bacterial populations [52]; for instance, bacterial communities on *Arabidopsis thaliana* leaves initially resemble airborne bacterial communities, and eventually transition to a distinct, phyllosphere-specific composition [53]. A dramatic turnover in phyllosphere bacteria on rice landraces suggests that plant-imposed selection favouring bacterial associates is either not very strong, or is not specific. Alternatively, the temporal variation may occur despite host selection; for instance, temporal changes in climate could alter the pool of *Methylobacterium* strains available for phyllosphere colonization, overcoming the impact of host selection. Yet another possibility is that variation in bacterial colonization may reflect variability in host selection: e.g. environmental factors may alter microhabitats in the rice phyllosphere, facilitating colonization by distinct bacteria across years. For instance, dispersal limitation and early stochasticity in colonization appear to play a major role during bacterial colonization of *A. thaliana* leaves [53]. Interestingly, of 13 distinct strains found in the air surrounding *A. thaliana* plants in the field, only one *Methylobacterium* strain typically colonized a given plant. In our case, 2017 was a drought year compared to 2016, such that rice plants were stunted and were presumably stressed. Prior work showed variable responses to drought and heat stress across rice cultivars, as manifested in flag leaf metabolites [35]. Moreover, sugars excreted on the leaf surface are heterogeneously distributed, leading to differentiation of bacterial population into sub-populations depending on access to sugars [54]. Hence, across years, bacteria could face context-dependent selection while colonizing rice leaves, potentially leading to variability in phenotype and community composition across years. To understand the patterns and causes of temporal variation in bacterial colonization, we need long-term monitoring of phyllosphere *Methylobacterium*, along with metadata on plant phenotype, stress levels, and key environmental variables.

Finally, we observed a curious lack of correlation between sugar availability in the phyllosphere and the ability of colonizing *Methylobacterium* strains to use those sugars. This pattern

begs the question: what processes generate and maintain variation in bacterial carbon use profiles? The variability is interesting because despite the predicted strong selection for methanol use [13], different rice landraces are colonized by bacteria with distinct abilities to use methanol as well as other carbon sources. Broadly speaking, such phenotypic variation could reflect evolutionary niche conservatism, local selection (e.g. for specialist or generalist phenotypes) [55], weak selection on specific metabolic pathways, or limited horizontal transfer of the relevant genes. To distinguish between these possibilities, we need long-term monitoring of the rice phyllosphere to determine spatial and temporal variation in airborne *Methylobacterium* communities (presumably the source of colonizing bacteria); leaf chemistry; and the identity, abundance, function, and impact of successful bacterial colonizers. Thus far, our results suggest that metabolic variation is not shaped strongly by evolutionary contingency (i.e. taxonomic relationships), or by direct host-imposed selection for carbon use. However, further work is necessary to validate our results, and to test the potential roles of selection and horizontal gene transfer in driving carbon use in *Methylobacterium*.

In summary, our work presents an important step towards understanding the diversity of natural populations of a widespread and dominant colonizer of the plant phyllosphere. Overall, our results demonstrate significant variability in rice-associated *Methylobacterium*, both at the level of community composition and metabolism. Our work also highlights the ecological and phenotypic diversity of *Methylobacterium* associated with multiple rice landraces within a small and biodiverse region of India, providing a platform to understand how bacterial communities can evolve with specific and divergent hosts in nature. Ultimately, we hope that such understanding will aid efforts to identify and tailor agriculturally important mutualisms that can support and preserve the cultivation of culturally and biologically important traditional rice landraces of NE India.

## Supporting information

**S1 Datasheet. Metadata for all *Methylobacterium* isolates sampled in September–October 2016 and 2017, along with the NCBI accessions used for analysis.**
(XLSX)

**S1 Table. List of distinct *Methylobacterium* isolates sampled in September–October 2016.** Sampling states are abbreviated as AR (Arunachal Pradesh) and MN (Manipur).
(DOCX)

**S2 Table. List of distinct *Methylobacterium* isolates sampled from seven focal landraces of Manipur in September 2017.** All abbreviations used for landraces are given in brackets after the landrace name.
(DOCX)

**S3 Table. List of *Methylobacterium* isolates sampled from landraces, seeds, soil, grasses, and commercial rice cultivated at the Chingarel field site, Manipur.**
(DOCX)

**S4 Table. List of distinct *Methylobacterium* isolates sampled from the seeds of landraces from Arunachal Pradesh (AR) and Manipur (MN).**
(DOCX)

**S5 Table. Repeatability of carbon utilization profile of *Methylobacterium* isolates, tested for a random subset of isolates across two experimental blocks.**
(DOCX)

**S1 Fig. Isolation of *Methylobacterium* sp. in the field using a customized portable enclosure that can be UV-sterilized.**
(TIF)

**S2 Fig. Correlation between geographic distance and phenotypic distance between isolates.** Scatter plot showing pairwise phenotypic distance vs. geographic distance for 91 distinct *Methylobacterium* isolates.
(TIF)

**S3 Fig. Concentration-dependent growth of *Methylobacterium* isolates on different carbon sources.** (A) A summary of the growth of *Methylobacterium* (58 isolates) on different concentrations of fructose, glucose, xylose, sucrose, and methanol. (B) Fraction of *Methylobacterium* showing different patterns of growth on each carbon source (see key for details). (C) Canonical variance analysis biplots of carbon use profiles, showing the clustering of *Methylobacterium* isolates by sampling state, elevation, and *Methylobacterium* clade.
(TIF)

**S4 Fig. Density of *Methylobacterium* colonies obtained from various sources.** Samples were imprinted or dilution-plated on Hypho minimal agar plates with 120mM methanol as the sole carbon source. Pink colonies are typical of *Methylobacterium sp*.
(TIF)

**S5 Fig. *Methylobacterium* associated with the phyllosphere vs. seed surface of rice landraces.** Each panel shows a pairwise comparison of the *Methylobacterium* community associated with phyllosphere vs. seed for a given each landraces, both in terms of community composition (piecharts) and carbon use phenotype (heatmap). (A) Landraces sampled in Arunachal Pradesh; (B) Landraces sampled in Manipur.
(TIF)

**S6 Fig. Phylogenetic relationships between rice-associated *Methylobacterium* isolates across years.** Neighbor-joining tree constructed from the 16S rRNA gene sequences of distinct *Methylobacterium* isolates from rice landraces (colored by sampling year), and closely related reference strains (in bold). Bootstrap values ≥ 50% are indicated. Clades are indicated on the right (see Fig 2).
(TIF)

**S7 Fig. Temporal variation in the composition and carbon use profiles of rice-associated *Methylobacterium* isolates.** Each panel shows a pairwise comparison between bacterial isolates from a given landrace sampled in 2016 vs. 2017 from Manipur, both in terms of community composition (piecharts) and carbon use profile (heatmap).
(TIF)

**S8 Fig. Sugars detected on rice leaf surfaces.** (A) GCMS chromatograms of standard sugars. (B) Representative chromatograms of flag leaf exudates of two landraces: Gegong (Arunachal Pradesh) and Chakhao (Manipur).
(TIF)

## Acknowledgments

We thank members of the Agashe lab for critical comments and discussion; Krushnamegh Kunte, Jharna Chakravorty, Kar Singh Megu, W Ibobi Singh, Indrajit, S Chandrakanta Singh, S Akeshore Singh and local farmers for assistance with field work; Sagolsem Surmangal Singh (Krishi Vigyan Kendra Thoubal, Manipur) for providing rice samples; IBSD (Institute of

Bioresources and Sustainable Development, Imphal) for access to laboratory facilities during fieldwork; Rittik Deb and Saurabh Mahajan for help with statistical analysis; Christopher Marx for kindly providing *M. radiotolerans* JCM strain; and the Korean Agriculture Culture Collection for providing reference strains KACC11585, KACC11761, KACC11437, KACC12263, KACC11431, KACC11543, and KACC11544. We also thank Mahesh Sankaran for access to a flatbed scanner, and the instrumentation and mechanical workshop teams at NCBS for making a portable enclosure for bacterial work. We acknowledge funding and support from the Department of Biotechnology (grant DBT-358 NER/AGRI/24/2013) and the National Centre for Biological Sciences.

## Author Contributions

**Conceptualization:** Pratibha Sanjenbam, Radhika Venkatesan, P. V. Shivaprasad, Deepa Agashe.

**Data curation:** Pratibha Sanjenbam.

**Formal analysis:** Pratibha Sanjenbam, Radhika Buddidathi, Radhika Venkatesan.

**Funding acquisition:** Deepa Agashe.

**Investigation:** Pratibha Sanjenbam, Radhika Buddidathi.

**Methodology:** Pratibha Sanjenbam, Radhika Venkatesan, Deepa Agashe.

**Project administration:** Deepa Agashe.

**Resources:** Radhika Venkatesan, P. V. Shivaprasad, Deepa Agashe.

**Supervision:** Radhika Venkatesan, P. V. Shivaprasad, Deepa Agashe.

**Validation:** Radhika Buddidathi.

**Visualization:** Pratibha Sanjenbam, Deepa Agashe.

**Writing – original draft:** Pratibha Sanjenbam.

**Writing – review & editing:** Radhika Venkatesan, P. V. Shivaprasad, Deepa Agashe.

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
