## [Decision Letter · Decision Letter 0]

2 Dec 2019

PONE-D-19-30230

Phenotypic Diversity Of Methylobacterium Associated With Rice Landraces In Northeast India

PLOS ONE

Dear Dr. Agashe,

Thank you for submitting your manuscript to PLOS ONE. After careful consideration, we feel that it has merit but does not fully meet PLOS ONE’s publication criteria as it currently stands. Therefore, we invite you to submit a revised version of the manuscript that addresses the points raised during the review process.

ACADEMIC EDITOR: 

Authors are advised to revise their work following the comments of the reviewers.

We would appreciate receiving your revised manuscript by Jan 16 2020 11:59PM. To enhance the reproducibility of your results, we recommend that if applicable you deposit your laboratory protocols in protocols.io, where a protocol can be assigned its own identifier (DOI) such that it can be cited independently in the future. For instructions see: http://journals.plos.org/plosone/s/submission-guidelines#loc-laboratory-protocols

We look forward to receiving your revised manuscript.

Kind regards,

Manoj Prasad, PhD

Academic Editor

PLOS ONE

Journal Requirements:

1. We note that you have stated that you will provide repository information for your data at acceptance. Should your manuscript be accepted for publication, we will hold it until you provide the relevant accession numbers or DOIs necessary to access your data. If you wish to make changes to your Data Availability statement, please describe these changes in your cover letter and we will update your Data Availability statement to reflect the information you provide.

Reviewers' comments:

Reviewer's Responses to Questions

**Comments to the Author**

1. Is the manuscript technically sound, and do the data support the conclusions?

Reviewer #1: Partly

Reviewer #2: Partly

2. Has the statistical analysis been performed appropriately and rigorously? 

Reviewer #1: Yes

Reviewer #2: Yes

3. Have the authors made all data underlying the findings in their manuscript fully available?

Reviewer #1: Yes

Reviewer #2: Yes

4. Is the manuscript presented in an intelligible fashion and written in standard English?

Reviewer #1: Yes

Reviewer #2: Yes

5. Review Comments to the Author

Reviewer #1: The author and the team did very good work in the field, lab, and compilation. However, few things need to be improved.

1. Northeast India is a vast area comprising of many states. Why only two states (Manipur and Arunachal Pradesh) were chosen for sample collection? The states of Assam and Tripura are major rice-producing states of the region and more Methylobacterium strains could have been isolated for better resolution.

2. The use of “landraces” as against “cultivars” needs to be justified.

3. Line no. 301-306: “We note two caveats here: (1) The 16S rRNA sequence does not provide sufficient resolution to identify.......”. Can this be solved by increasing the population size (refer to query 1)? If not how?

4. Why only Carbon use profile was selected as a parameter for bacterial colonization?

5. Landraces were found to be a good choice for Methylobacterium colonization however strain specificity could not be established. And considering the case of Chakhao (Manipur) where temporal variation was least observed (for two-year study), does aromatic character of the cultivar had a role to play?

6. Does Methalobacterium colonization have a negative impact on rice physiology or host deficit or plant defense?

Reviewer #2: Appreciate the author and team for working on such important topic but the current manuscript need revision in terms of scientific language. 

6. PLOS authors have the option to publish the peer review history of their article (what does this mean?). If published, this will include your full peer review and any attached files.

Reviewer #1: No

Reviewer #2: Yes: Avinash Chandra Pandey

---

## [Author Response · Author response to Decision Letter 0]

5 Dec 2019

Please see our detailed response to reviewers in the file uploaded with the revised manuscript.

---

## [Decision Letter · Decision Letter 1]

21 Jan 2020

Phenotypic Diversity Of Methylobacterium Associated With Rice Landraces In Northeast India

PONE-D-19-30230R1

Dear Dr. Agashe,

We are pleased to inform you that your manuscript has been judged scientifically suitable for publication and will be formally accepted for publication once it complies with all outstanding technical requirements.

With kind regards,

Manoj Prasad, PhD

Academic Editor

PLOS ONE

Additional Editor Comments (optional):

Reviewers' comments:

Reviewer's Responses to Questions

**Comments to the Author**

1. If the authors have adequately addressed your comments raised in a previous round of review and you feel that this manuscript is now acceptable for publication, you may indicate that here to bypass the “Comments to the Author” section, enter your conflict of interest statement in the “Confidential to Editor” section, and submit your "Accept" recommendation.

Reviewer #1: All comments have been addressed

Reviewer #2: (No Response)

2. Is the manuscript technically sound, and do the data support the conclusions?

Reviewer #1: Yes

Reviewer #2: Yes

3. Has the statistical analysis been performed appropriately and rigorously? 

Reviewer #1: N/A

Reviewer #2: Yes

4. Have the authors made all data underlying the findings in their manuscript fully available?

Reviewer #1: Yes

Reviewer #2: Yes

5. Is the manuscript presented in an intelligible fashion and written in standard English?

Reviewer #1: Yes

Reviewer #2: Yes

6. Review Comments to the Author

Reviewer #1: The authors have addressed all the issues with relevant references wherever necessary. Overall I think the author's responses are reasonable and this manuscript should be evaluated in a full review process.

Reviewer #2: It was unpleasant that Author and team have not addressed the reviewer suggestions in previous reviewer submitted file. Therefor reviewer resubmitting the same file. Please find the attached file. Hope an improved version of manuscript.

7. PLOS authors have the option to publish the peer review history of their article (what does this mean?). If published, this will include your full peer review and any attached files.

Reviewer #1: No

Reviewer #2: Yes: Avinash Chandra Pandey

---

## [Editor Report · Acceptance letter]

12 Feb 2020

PONE-D-19-30230R1 

Phenotypic Diversity Of Methylobacterium Associated With Rice Landraces In Northeast India 

Dear Dr. Agashe:

I am pleased to inform you that your manuscript has been deemed suitable for publication in PLOS ONE. Congratulations! Your manuscript is now with our production department. 

With kind regards,

on behalf of

Dr. Manoj Prasad 

Academic Editor

PLOS ONE